# Aptamers: Potential Diagnostic and Therapeutic Agents for Blood Diseases

**DOI:** 10.3390/molecules27020383

**Published:** 2022-01-07

**Authors:** Maher M. Aljohani, Dana Cialla-May, Jürgen Popp, Raja Chinnappan, Khaled Al-Kattan, Mohammed Zourob

**Affiliations:** 1Institute of Physical Chemistry and Abbe Center of Photonics, Friedrich Schiller University, Helmholtzweg 4, 07743 Jena, Germany; dana.cialla-may@leibniz-ipht.de (D.C.-M.); juergen.popp@ipht-jena.de (J.P.); 2Department of Pathology, College of Medicine, Taibah University, Madinah 42353, Saudi Arabia; 3Leibniz Institute of Photonic Technology, Albert-Einstein-Str. 9, 07745 Jena, Germany; 4Center for Applied Research, InfectoGnostics Research Campus Jena, University of Jena, Philosophenweg 7, 07743 Jena, Germany; 5Department of Chemistry, Alfaisal University, Riyadh 11533, Saudi Arabia; rchinnappan@alfaisal.edu; 6College of Medicine, Alfaisal University, Al Zahrawi Street, Al Maather, Al Takhassusi Rd, Riyadh 11533, Saudi Arabia; KKattan@alfaisal.edu

**Keywords:** aptamers, diagnostic, therapeutic, blood diseases

## Abstract

Aptamers are RNA/DNA oligonucleotide molecules that specifically bind to a targeted complementary molecule. As potential recognition elements with promising diagnostic and therapeutic applications, aptamers, such as monoclonal antibodies, could provide many treatment and diagnostic options for blood diseases. Aptamers present several superior features over antibodies, including a simple in vitro selection and production, ease of modification and conjugation, high stability, and low immunogenicity. Emerging as promising alternatives to antibodies, aptamers could overcome the present limitations of monoclonal antibody therapy to provide novel diagnostic, therapeutic, and preventive treatments for blood diseases. Researchers in several biomedical areas, such as biomarker detection, diagnosis, imaging, and targeted therapy, have widely investigated aptamers, and several aptamers have been developed over the past two decades. One of these is the pegaptanib sodium injection, an aptamer-based therapeutic that functions as an anti-angiogenic medicine, and it is the first aptamer approved by the U.S. Food and Drug Administration (FDA) for therapeutic use. Several other aptamers are now in clinical trials. In this review, we highlight the current state of aptamers in the clinical trial program and introduce some promising aptamers currently in pre-clinical development for blood diseases.

## 1. Introduction

Nucleic acid aptamers constitute a special class of synthetic polymers or oligomers of single-stranded ssDNA or RNA molecules, and they have the capacity to bind to a specific target by forming secondary and/or tertiary structures. The word “aptamer” derives from the Latin word *aptus,* meaning “to fit”, and the Greek word *meros,* meaning “particle”, and was chosen to describe the “lock and key model” of the relationship between aptamers and their binding targets. Aptamers bind with a high affinity and specificity to a wide range of targets, such as proteins, peptides, small molecules, metal ions, bacteria, viruses, and whole live cells. 

Aptamers were first developed in 1990 during an experiment by Tuerk and Gold using the systematic evolution of ligands in an exponential enrichment (SELEX) procedure [1]. The SELEX procedure was used for the aptamer selection process, and in a typical SELEX, it was carried out using purified target molecules, starting with a large library of random oligonucleotides. The oligonucleotides that were strong binders to the target molecule were selected from the initial library through cycles of target binding, selection, and amplification (Figure 1).

Aptamers can be considered a promising class of molecules that are the chemical equivalents of antibodies. Monoclonal antibodies (mAbs) are recognized as one of the most powerful tools in modern medicine for therapeutic and diagnostic applications. Although aptamers are comparable to traditional antibodies, they possess some superior aspects, including a high chemical stability and quick and easy bulk chemical production. In addition, they can be produced on a large scale with low cost, while retaining high reproducibility and reliability (Table 1). 

Aptamers are produced using animal-free technologies and offer a superior alternative to mAbs (Figure 2). 

Aptamers can bind to highly toxic or non-immunogenic antigens, an attribute that cannot be achieved with animal-based methods of mAb production. Aptamers are intermediate in size (8–15 kDa) between antibodies (150 kDa) and small peptides (1–5 kDa), and about 20 times smaller than antibodies [3]. Their small size can lead to better tissue penetration, which can be used in solid tumor therapy.

Pegaptanib, an aptamer-based therapy developed from the NX1838 aptamer, serves as an anti-angiogenic agent and is selective for the vascular endothelial growth factor (VEGF). Pegaptanib can efficiently bind and inhibit VEGF to limit VEGF–cellular interactions. It was the first aptamer to receive U.S. Food and Drug Administration (FDA) approval for the treatment of ocular vascular disease and is now in several clinical trials as a therapeutic. The success of pegaptanib provides ample evidence of the potential for using aptamers’ specific and reversible target binding and inhibition as a promising strategy for new therapies [4].

The use of aptamers has been widely investigated in several biomedical areas, such as biomarker detection, diagnosis, imaging, and targeted therapy. Currently, the aptamers used in cancer therapy are divided into free RNA and DNA aptamers and are specific to molecular targets that are the hallmarks of diseases. Because aptamer-based therapeutic properties are similar to those of mAbs, aptamers can be described simply as chemical antibodies. They are capable of binding and inhibiting the immunoregulatory components of carcinogenesis, and they can also be used as carriers for therapeutic agents. Aptamer bind drugs through covalent and non-covalent conjugation methods [5]. Over the last few years, aptamers have emerged as promising alternatives to antibodies and have been used in biosensing, disease diagnosis, therapeutic applications, and a wide range of other applications. In this review article, we focus on aptamer applications in the diagnosis and therapy of hematological diseases [6,7].

### 1.1. Aptamer Selection Technology

The SELEX process is an in vitro selection method for isolating DNA or RNA sequences that bind to a specific target. It involves selecting RNA/DNA ligands (aptamers) of oligonucleotides that can be used to screen and select, with a high affinity, a wide range of target molecules from random libraries of RNA/DNA [4]. The SELEX process is the gold standard for selecting aptamers. Independent of ssDNA or RNA sequences with purified proteins, small molecules, metals, whole-cell, or living organisms, the protocols for a conventional aptamer selection involve three important technical steps: (i) incubation of a target molecule with the random sequenced ssDNA/RNA library, (ii) separation of the target bound sequences from the non-bound sequences, and (iii) recovering the bound sequences, followed by the amplification of the bound sequences. The SELEX cycles are repeated until the aptamer sequence reaches a significantly high affinity. Because the SELEX process is complex, it usually requires several weeks to complete. Although the basic steps of the aptamer selection process do not change, new technologies for SELEX have provided opportunities to enhance and accelerate. Recent developments have also helped to minimize the effort, time, and cost of aptamer selection, thereby overcoming technical difficulties and improving the success rate of aptamer screening. The advanced technologies enhance the SELEX protocol for developing new aptamers for the desired targets [8].

During the SELEX process, the target molecules are immobilized on a solid support (e.g., chromatographic beads or sepharose beads) to facilitate the separation of target-binding oligonucleotides from non-binding ones. The ssDNA library pool of 10^15^ random nucleotide sequences are incubated with immobilized target molecule and the enrichment in the ssDNA binding aptamer is monitored by the fluorescence [9]. SELEX can be performed against the purified protein as well as against whole cells by cell-SELEX, and the ssDNA/RNA aptamers bind to specific cell membrane proteins existing on live cells. Aptamers selected from cell-SELEX can identify target cancer cells and be used for cancer diagnostics. Furthermore, high-affinity and specific aptamers can be generated by cell-SELEX against live pathogenic organisms, such as bacteria and viruses [10,11].

It is crucial to consider that the aptamers selected from the conventional SELEX method using purified proteins might not be applied for a real sample analysis. Using purified proteins as targets offers the advantage of easy enrichments and a high affinity and specificity if the protein has the stable conformation as in the native (in vivo) condition. There are many clinically important proteins, such as purified MUC1 peptides [12], the purified extracellular domain of the prostate-specific membrane antigen (PSMA) [13], the cell adhesion molecule P-selectin [14], and the protein tyrosine phosphatase 1B [15] which have been used as targets. If the target protein is in different conformations of the aptamer binding pocket, it is masked or blocked by other associated biomolecules in the physiological condition, and the aptamer might not recognize the target proteins.

The in vitro selection of an RNA aptamer against histidine-tagged EGFRvIII ectodomain has been performed using an *Escherichia coli* system for protein expression and purification. The aptamer had a high affinity (K_d_ = 33 nM) and specificity for the target. However, it did not bind to the same protein expressed from eukaryotic cells, because of the post-translational modification (glycosylation) of EGFRvIII, which has a different structural conformation from the protein used for aptamer selection [16]. Cell-SELEX is a complex process that involves technical challenges [17]. To select aptamers for target proteins that are expressed less on the cell surface, the cell surface has to be accessible for the aptamer library to bind to the healthy and viable cell lines and to minimize the nonspecific aptamer during the SELEX process [18]. Dead cells can be removed using a fluorescence-activated cell sorter (FACS) and microbead-based methods [19,20]. In general, cell-SELEX requires more rounds of selection to develop high-affinity aptamers than the conventional SELEX methods [21,22].

Several strategies have been introduced to modify and optimize SELEX methods. For example, counter-SELEX improves aptamer selection by discarding nonspecific aptamers, and this process is conducted by adding a pre-clearing step to the SELEX sequence that uses closely related structural analogs of the target. Another approach is using capillary electrophoresis-SELEX to separate a bound target from unbound nucleic acids. Because nucleic acids (aptamers) that bind the target have different mobilities, they can be collected as separate fractions. A new electrochemical approach (electrochemical SELEX), based on immobilizing the target analyte on gold electrodes, has also been used. This approach eliminates the use of beads as a solid support matrix and the fluorescent labels used in SELEX [23]. Another example is cell-SELEX, a unique methodology for the in vitro selection of aptamers for whole-cell. This form of SELEX is not limited to individual molecules but targets whole living cells, such as cancer cells. This approach generates cell-specific aptamers and can identify unknown biomarkers for cancer cells [8,10,11,24].

### 1.2. Aptamer Optimization and Modification

Over the last few decades, rapid advancements in aptamer development and nanotechnology have rendered aptamers an attractive tool for biomedical applications [25,26]. They can be chemically modified very easily, and several modifications which are achieved easily without compromising aptamer-target interactions. Introducing such changes to aptamers can be accomplished during or after the SELEX process. Generally, aptamers are susceptible to nuclease degradation and fast renal excretion, which significantly limits their in vivo applications [27,28], and numerous attempts have been undergone to overcome these obstacles. Aptamer modification can enhance the binding affinity with the target, improve stability, and avoid degradation by in vivo nucleases [29,30]. Modifying aptamers for biological applications and enhancing their in vivo stability and pharmacokinetics in biological environments can be achieved by adjusting the SELEX protocol to optimize the aptamer libraries and separation schemes and identify aptamer sequences from enriched libraries, aptamers can be modified for biological applications and to enhance their in vivo stability and pharmacokinetics in biological environments.

Because aptamers are single-stranded nucleic acids, unchanged aptamers are unstable in biological fluids and have a short half-life caused by enzymatic degradation in serum and body fluids. Many new polymerases are available for producing libraries of more stable aptamers, protected from the nuclease hydrolysis; thus, avoiding post-selection modification. Introducing unnatural nucleotides into the library would lead to more stability in the presence of nucleases, and this process can be achieved by altering the sugar rings including 2′-fluoro (2′-F) ribose, 2′-amino (2′-NH2) ribose, 2′-O-methyl (2′-OMe) ribose, and locked nucleic acids (LNAs), (bridging the 2′- and 4′-ribose positions covalently) [31,32]. SomaLogic has introduced the chemical modification of aptamers, in which chemical modification is carried out on the bases for the stability of the aptamers in the presence of biological fluids, enhancing the structural diversity and strong target binding ability. The modified aptamers (SOMAmers) have reflected their best biological functions in terms of their stability and affinity compared to conventional aptamers. Researchers have developed SOMAmers by using heterocycles, hydrophobic groups, phenyl, large naphthyl, and a more complicated indole to replace the dT base in the DNA library with dU modified at the 5′ position of the base [33]. Many diverse SOMAmer-based array techniques, such as SOMAscan and SOMApanel, have been used for clinical applications [34]. Phosphate linkage modifications have been introduced in the nucleic acid backbone to improve the stability and binding ability by substituting sulfur-containing ester bonds for conventional phosphate bonds [35]. Constructing RNA origami, a nanoscale folding of RNA enables the production of RNA origami anticoagulants [36]. The Spiegelmer is a new class of drug that consists of mirror-image aptamers. The word *Spiegelmer* derives from the German word *spiegel,* meaning “mirror.” Spiegelmers are aptamers composed of natural D-oligonucleotides, which can be selected against mirror-image targets, such as D-amino acid peptides, rather than natural L-amino acid peptides. The L-form RNA is nuclease resistant and suitable for in vivo applications because of its improved aptamer stability [37]. Nuclease-resistant circular aptamers are used to achieve the metabolic stability of aptamers in the presence of serum or in the biological fluids. The ligation of 5′ and 3′ terminals in a nucleic acid is protected from the exonuclease degradation. A stable anticoagulant multivalent circular aptamer has developed. The cyclization increases the thermal stability which keeps all the aptamers in a uniform conformation. The modification of oligonucleotides with cholesterol leads to a highly resistant form of nuclease hydrolysis in the serum. A modified aptamer has a several-fold longer half lifetime compared to the unmodified one.

### 1.3. Aptamers for Blood Diseases

Knowledge of cancer cells and the molecular mechanisms of their diseases have rapidly expanded in the last decade, which helped to identify the defects and limitations of conventional strategies for diagnosing and treating hematological diseases [38,39]. Nanotechnology, including the use of aptamers, now provides transformative tools to translate biomedical findings into novel diagnostic, therapeutic, and preventive tools to treat different types of diseases in hematology [40,41].

This review discusses the progress of research on aptamers and their use in blood diseases according to literature published in recent years [39]. It provides a summary of the current work and a broad perspective on the hematological applications of aptamers, and it also outlines the principles of these hematological applications to provide an insight into their therapeutic successes and failures (Table 2).

## 2. Aptamers for Hematologic Oncology

### 2.1. Anti-Nucleolin Aptamers for Acute Myeloid Leukemia (AML) Treatment

AML is a type of blood cancer from the myeloid line of hematopoietic cells, and it is a heterogeneous disease with poor survival and a high risk of a relapse. Nucleolin is a major multifunctional nucleolar protein involved in regulating transcription events, including various cell proliferation and growth aspects. Nucleolin is expressed more significantly on several cancer cells than in normal cells and seems to promote tumor growth. In hematology, nucleolin’s enforced expression increases leukemia cell proliferation and affects both the pathogenesis and prognosis of AML. Because nucleolin is a membrane protein that serves as a binding receptor for various ligands involved in cancer pathogenesis, it represents a potential strategic target for cancer therapy and an attractive target for several types of malignancies, including AML [79,80].

AS1411 is the first aptamer-based anticancer therapeutic that has undergone human clinical trials. It is a unique DNA aptamer that consists of a 26-base nucleolin-binding G-rich oligonucleotide, and it uses a novel approach for targeting nucleolin. Initially named AGRO100, it was renamed to AS1411 and then ACT-GRO-777. AS1411 combined with high-dose cytarabine is used as a cancer-targeting agent. The combination of AS1411 and high-dose cytarabine has been evaluated in phase II clinical trials for refractory and relapsed AML (ClinicalTrials.gov, #NCT00512083 and #NCT01034410), and the results were promising. This combination had a synergistic effect on cancer cell growth inhibition and an acceptable safety profile with side effects typically associated with cytarabine treatment in these patients [42,81,82]. Nanomaterials constructed using the AS1411 aptamer have been successfully applied for the inhibition of tumor growth [83,84,85].

### 2.2. CD33-Specific Aptamer for AML Treatment

CD33 is a transmembrane protein that is expressed by mature myeloid cells, AML blasts, and normal myeloid progenitors [86]. The recent use of a humanized anti-CD33 mAb in combination with chemotherapy has been considered a major advance against AML. Actinium-225 (225Ac, alpha emitter isotopes that emit α-particles)-lintuzumab is a new approach in AML therapy; it is composed of actinium-225 linked to a humanized anti-CD33 mAb and a low-dose cytarabine (LDAC). This drug can safely induce remission in older patients with untreated AML [87]. CD33-specific aptamers have properties comparable to anti-CD33 antibodies in terms of binding and internalization into CD33-positive myeloid cell lines, and they also have the potential to carry chemotherapeutic drugs to CD33-positive cells in adult and pediatric patients with AML [43]. The CD-33 aptamer was conjugated with doxorubicin (Dox) and produces Dox–aptamer conjugates. The target-specific drug aptamer could inhibit CD33-positive acute myeloid leukemia [88].

### 2.3. Anti-CD30 Aptamers for Diagnosing and Treating CD30-Positive Malignant Lymphomas

CD30 is a cell membrane receptor that is significantly expressed in some types of lymphoma cells in classical Hodgkin’s lymphoma; it is also expressed by a subset of diffuse large B-cell lymphoma (DLBCL) cells [44]. CD30 is used as a therapeutic target, and the ssDNA aptamer for CD30 selected via a hybrid SELEX methodology specifically binds to the CD30 receptor targets with a high affinity. This aptamer has been modified to the truncated variant of the CD30 aptamer, which has a 50-fold higher affinity than its longer version. The CD30 aptamer works by inducing the oligomerization of CD30 receptors and, ultimately, inducing the apoptosis of lymphoma (ALCL) cells. This aptamer-based model of immunotherapy offers an alternative to targeted mAbs and has the potential to transform the lymphoma treatment [45].

CD30 aptamers are potential agents for disease diagnosis. A fluorescently labeled RNA aptamer was tested in cultured anaplastic large cell lymphoma and Hodgkin’s lymphoma cells that expressed high levels of CD30. The flow cytometry and fluorescence microscopy revealed the specific and sensitive binding of a CD30 aptamer probe at low concentrations (0.3 nM) of CD30-positive lymphoma cells. The CD30 aptamer-based probe shows a potential application in the multicolor flow cytometry for detecting CD30-positive cells, indicating that it can act as an alternative or supplement to antibodies for diagnosing CD30-positive lymphomas [89]. Anti CD30 aptamer-conjugated nanoparticles are a potential candidate for the specific delivery of doxorubicin to anaplastic large cell lymphoma cells [90].

### 2.4. BAFF-R-Specific Aptamer for Non-Hodgkin’s Lymphoma (NHL)

The B-cell activating factor receptor (BAFF-R) is necessary for B-cell maturation and survival. A high expression of BAFF receptors has been recognized in several B-cell malignancies (e.g., follicular lymphoma, DLBCL, mantle cell lymphoma, and Burkitt’s lymphoma) but not in T/NK cell lymphoma or Hodgkin’s lymphoma [46]. Aptamers for BAFF-R that are expressed by B-cell lymphoma cells are developed by cell-SELEX, and this aptamer binds specifically to BAFF-R. Zhou’s group isolated 20-fluoro-modified RNA aptamers that bind specifically to BAFF-R from an 81-nucleotide RNA library using in vitro SELEX. The BAFF receptor aptamers showed specific binding and internalization in the BAFF-R-positive lymphoma cells but, not in the BAFF-R-negative T-cells (CEM) [47]. The B-cell-specific aptamer was labelled with Cy5 and used for fluorescence imaging in tumor xenograft nude mice, which was used for the temporal mapping of the aptamer. The Cy-5 aptamer successfully recognized the tumor cells [91,92].

### 2.5. Anti-CXCL12 Spiegelmer in Chronic Lymphocytic Leukemia and Multiple Myeloma

In cancer, the tumor microenvironment and interactions between tumor cells and the cellular and noncellular components of the tumor microenvironment play a critical role in the initiating, maintaining, and relapsing of hematological and solid tumors. The CXCR4/CXCL12 axis plays a crucial role in multiple myeloma (MM) cells homing to the bone marrow and the interaction between the bone marrow microenvironment and MM cells. CXCL12 (stromal cell-derived factor-1/SDF-1) is a chemokine that binds to membrane proteins, specifically the CXC receptor 4 (CXCR4, CD184) and CXC receptor 7 (CXCR7) [93], and it is expressed by stromal cells in several tissues and organs, including bone marrow [94,95,96]. The interaction of cancer cells with the surrounding microenvironment plays a critical role in chronic lymphocytic leukemia (CLL), and MM cells utilize the CXCR4/CXCL12 axis for bone marrow homing. In MM, SDF-1 is an essential participant of the bone marrow microenvironment, regulating numerous processes related to its malignant transformation during MM development. CXCL12 also plays a crucial role in chemoresistance via leukemia–stromal interactions. The homing of CLL cells into the bone marrow and lymph node’s protective microenvironments rescues these tumor cells from both spontaneous and chemotherapy-induced apoptosis [49].

A new therapeutic approach of targeted therapy can attack cancer cells through the SDF-1/CXCR4 axis, which is a fundamental driving force for tumor cell growth and survival. The NOX-A12 (olaptesed pegol) Spiegelmer is a pegylated L-stereo-isomer RNA aptamer (a mirror-image RNA oligonucleotide) that binds and neutralizes CXCL12 and CXCL12 (anti-CXCL12), and it forms the basis of the therapeutic concept. The anti-CXCL12 aptamer can inhibit tumor-supporting pathways and mobilize CLL cells away from their protective microenvironment, thereby inducing apoptosis and chemotherapy sensitization in these leukemic cells [48]. In MM, CXCL12 serves as a chemokine that regulates various MM development processes through signaling via CXCR4 and CXCR7. In a phase II pilot study, to evaluate the activity and safety of olaptesed, this aptamer was combined with BTZ and DEX for patients with relapsed or refractory MM. The results showed that this combination was safe and well-tolerated without any significant increase in adverse events, and a single dose of olaptesed effectively mobilized MM cells. A CXCR4 antagonist is another way to target the SDF-1/CXCR4 axis for cancer treatments with an overexpression of the SDF-1/CXCR4 axis such as in MM. This approach is a promising strategy for disrupting myeloma–stroma interactions and inhibiting myeloma growth and survival by disrupting the adhesion of MM cells to bone marrow stromal cells [50,51,52].

### 2.6. CD38-Specific Aptamer in Multiple Myeloma

CD38 is a transmembrane glycoprotein that has been broadly used to recognize plasma cells and MM, and it is considered an attractive biomarker for targeted MM therapy. In a study examining targeted therapy for MM, researchers used CD38-specific ssDNA aptamers selected by a hybrid SELEX process to efficiently target myeloma cells. The aptamers were conjugated to cytotoxic agent doxorubicin (DOX), forming an aptamer–drug conjugate (ApDC) to target and release this agent within MM cells. Subsequently, ApDCs induced MM cell death by apoptosis, but they did not affect CD38-negative cells and had only a minimal impact on control cells [53].

### 2.7. Aptamers for B-Cell Burkitt’s Lymphoma Cells

High-quality aptamers were developed using cell-SELEX for a viable B-cell Burkitt’s lymphoma cell line (Ramos cells) as the target. Among a panel of DNA-selected aptamers, the TD05 aptamer was recognized for its specificity to target proteins on the surface of Ramos cells. TD05 is an immunoglobulin-heavy mu chain (IGHM) that is not present in normal CD19+ B cells or other hematopoietic cells. Because IGHM is the main component of the B-cell receptor complex, it can be used to identify new potential targets for the therapeutic regulation of the neoplastic B-cell function [54].

The successful use of Cy5-labeled aptamer TD05 (Cy5-TD05) as a probe for in vivo aptamer-based molecular imaging in Ramos cells confirmed that these aptamers can recognize tumor cells with a high sensitivity and specificity. Although fluorescence-labeled aptamers are promising molecular probes for cancer diagnostics and in vivo imaging [91], the TD05 aptamer has poor stability in blood, and the degradation by nucleases can affect the use of aptamer-based in vivo probes for cancer diagnosis. A polyethylenimine (PEI)–aptamer complex was generated for in vivo cancer imaging by using deoxyribonuclease (DNase)-activatable fluorescence probes (DFProbes) to follow the degradation of DNA aptamer. PEI-protected aptamer molecular probes were used successfully to protect the TD05 aptamer probes from DNase degradation while still maintaining this aptamer’s ability to recognize Ramos cells [91].

### 2.8. Aptamers for T-Cell Acute Lymphoblastic Leukemia (T-ALL) Cell Lines

Initially, a number of DNA aptamers were selected through a cell-SELEX selection process for the recognition of human T-cell ALL CCRF-CEM cell lines. The sgc8 aptamer was then selected to identify the human protein tyrosine kinase-7 (PTK7) as a binding target protein present on the leukemia cell surface [55]. Sgc8 is a DNA aptamer that can specifically recognize human T-cell ALL CCRF-CEM cell lines, and it is specific for the membrane receptor and PTK7 as a molecular target [55]. This two-step strategy for aptamer selection is very effective in biomarker discovery and requires no prior knowledge of the cell biomarker population. It is also valuable in identifying biomarkers for minimal residual disease (MRD) diagnostics in the detection of leukemia cells in bone marrow [56].

### 2.9. Aptamer–Drug Conjugates for Targeted Drug Delivery to Tumor Cells

Aptamer–drug conjugates are able to distinguish between target leukemia cells and normal human bone marrow aspirates. The sgc8 aptamer has been conjugated to the anticancer drug DOX for targeted drug delivery to PTK7 T-ALL leukemia cells. The sgc8 aptamer–DOX conjugate shows a high binding affinity and is efficiently internalized by the target cells [57]. In addition, sgc8-PEG-liposome nanoparticles provide powerful binding to the target cells and enhanced cellular internalization across the cell membrane [97]

### 2.10. CD117-Specific Aptamer in AML

CD117 (c-Kit) is a transmembrane receptor that is highly expressed in leukemia cells in 95% of patients with relapsed AML [60]. Because the ideal targeted therapy should be specific to AML cells with no effect on normal cells, CD117 is a possible molecule for developing a new targeted therapy [98]. Recently, hybrid SELEX using human erythroleukemia (HEL) cells expressing the CD117 antigen identified CD117-specific ssDNA aptamers with a high ability to bind to AML cells. The researchers used an aptamer–MTX conjugate to target CD117-positive AML cells for the selective growth inhibition of leukemia cells, and they found that it had no toxicity to normal cells. This result illustrates the potential clinical value of aptamers in AML treatment [61].

## 3. Aptamers in Hemostasis Disorders

### 3.1. Aptamers for von Willebrand Factor-Related Diseases

The ARC1779 aptamer is a pegylated form of the DNA aptamer ARC1172. The pegylation prevents digestion by nucleases while stabilizing the same affinity for the von Willebrand factor (VWF) A1 domain. Pegylation also inhibits the prothrombotic function of the VWF by blocking the binding of the VWF A1 domain to platelet GPIb receptors and by reducing platelet adhesion, aggregation, and thrombus formation. ARC1779 is an intravenous infusion agent for patients with VWF-related platelet function disorders of the thrombotic thrombocytopenia purpura (TTP) and von Willebrand disease type 2B [99]. The aptamer-based anti-von Willebrand factor (ARC1779) is a potent inhibitor of the VWF A1 domain interaction with GPIb and effectively prevents the consumption of VWF and platelets in response to desmopressin in VWD type 2B and prevents desmopressin-induced thrombocytopenia in VWD type 2B [62]. A clinical trial assessed the safety, pharmacokinetics, and pharmacodynamics of ARC1779 injection in patients with VWF-related platelet function disorders, and the results suggested that ARC1779 can inhibit platelet aggregation without a significant increase in bleeding in healthy volunteers [63]. A phase II clinical trial proposed to study the effect of ARC1779 on cerebral microembolism in patients undergoing carotid endarterectomy in the immediate postoperative period, and it planned to recruit 100 patients. However, the study had to be suspended after recruiting only 36 patients [64]. In a phase II clinical trial, the anti-VWF aptamer ARC1779 was used in patients with TTP. However, as mentioned earlier, the study was prematurely closed. Nonetheless, significant observations of the acute TTP patients enrolled confirmed that blocking the A1 domain of the VWF had the potential to increase platelet counts when used in combination with plasma-exchange therapy [100]. DTRI-031, an anti VWD aptamer, was selected, which selectively binds and inhibits the VWD mediated platelet adhesion and arterial thrombosis, while enabling the rapid reversal of this antiplatelet activity by an antidote oligonucleotide (AO). The dose-dependent study indicates the inhibition of platelet aggregation and thrombosis in whole blood and mice. The aptamer can achieve a potent vascular recanalization of platelet-rich thrombotic occlusions in murine and canine carotid arteries. A murine toxicological study of the aptamer showed that it is very well tolerated in the environment [101].

### 3.2. Aptamers in Hemophilia

The tissue factor (TF) is important in hemostasis and performs a key step in the initiation of the extrinsic (tissue factor) pathway of the blood coagulation cascade. TF is controlled by the tissue factor pathway inhibitor (TFPI), considered to be the primary inhibitor of beginning blood coagulation; it also modulates the severity of a variety of bleeding and clotting disorders [102]. The TFPI, or extrinsic pathway inhibitor is a natural anticoagulant synthesized by endothelial cells and megakaryocytes. It is produced from a single gene transcription of alternatively spliced mRNAs that translate into three principal spliced isoforms in humans: TFPIα, TFPIβ, and TFPIδ. TFPIα (or the full-length TFPI) is the predominant isoform expressed in humans, and these isoforms differ in their C-terminal domain structure, cell surface association mechanism, and anticoagulant activity. The TFPI is primarily distributed in endothelial cells, and a small fraction of body TFPI circulates in plasma or is bound to lipoproteins [103]. TFPI negatively regulates the coagulation cascade by inhibiting two potent procoagulant complexes. TFPI complexes with factor Xa subsequently impair the extrinsic pathway coagulation system’s trigger mechanism by the tissue factor and factor VIIa (TF–FVIIa) complex [65]. The anti-TFPI aptamer, BAX 499, (formerly ARC19499) binds to multiple domains: Kunitz 1, Kunitz 3, and the C-terminal domains of TFPI. The inhibition of TFPI enables the initiation and propagation of blood coagulation via the extrinsic pathway of the coagulation system for thrombin generation and clot formation [66]. Another thrombin-binding aptamer, HD1, inhibits the conversion of soluble fibrinogen into insoluble fibrin strands. However, it is easily degraded by nucleases in vivo. A combination of duplex/quadruplex sequences and homo and hetero-bivalent constructs have introduced the biological performances in therapeutic applications [104].

In hemophilia A, factor VIII deficiency limits clot propagation via the intrinsic pathway. Inhibiting TFPI by antagonists directs the coagulation process to a primitive condition that enables the initiation and propagation of the blood coagulation cascade and clot formation of the inhibitor, and TFPI enables this response via the extrinsic (TF) pathway and clot formation. Using an anti-TFPI aptamer (BAX 499) is a new treatment strategy for hemophilia by interfering with the TFPI inhibition of both factor Xa and the tissue factor/factor VIIa complex. This enables the initiation of TF-mediated coagulation and propagation via the extrinsic pathway for the treatment of bleeding associated with hemophilia [67]. The first-in-human and proof-of-mechanism study of an anti-TFPI aptamer was conducted in hemophilia patients to test the safety and tolerability of ARC19499. The administration involved single and multiple injections, which elevated the TFPI plasma levels; this response was due to the induction of the binding of intracellularly stored TFPI and the binding of BAX 499 to the Kunitz 3-C terminus domain of TFPI, thereby prolonging the circulatory half-life of full-length TFPI [68,69].

## 4. Aptamers for Hemoglobinopathies

### 4.1. Anti-P-Selectin RNA Aptamers for Sickle Cell Disease

Sickle cell disease (SCD) is the most common hereditary blood disease and can lead to severe complications, such as hemolytic anemia, episodic vaso-occlusion, and progressive multiple organ damage. The major pathophysiology of SCD complications are related to a poor microvascular blood flow and adhesive interactions between circulating sickle red blood cells (RBCs), leukocytes, and endothelial cells. A correlation exists between the clinical vaso-occlusive severity and adherence of the red blood cells (RBCs) to endothelial cells [70]. P-selectin, a cell adhesion molecule is expressed in activated vascular endothelial cells and platelets. One of the new therapeutic strategies for managing SCD is to target these major complications by inhibiting adhesive interactions with endothelial cells [71].

Crizanlizumab (formerly SelG1), an antibody-based P-selectin inhibitor used against the adhesion molecule P-selectin, was evaluated in a double-blind, randomized, placebo-controlled phase II clinical trial to assess its efficacy. Patients who received high-dose crizanlizumab had significantly fewer sickle cell-related pain crises per year compared with control groups who received the placebo [105]. In mice, an aptamer-based P-selectin inhibitor was developed and investigated, and it showed the ability to inhibit the adhesion of sickle RBCs and leukocytes to endothelial cells by 90% and 80%, respectively, thereby showing potential as a new therapeutic agent for SCD [106].

### 4.2. Aptamers for Complement-Related Disorders

The complement system makes up part of the immune system and is composed of several distinct plasma proteins. These complements play a crucial role in the immediate responses for protection from common pathogens. Complements are groups of plasma proteins that play an important role in innate and acquired host defense mechanisms against infection and in various immunoregulatory processes. Usually, complements are inactive until stimulated by recognizing exogenous materials or pathogen-associated molecules, after which an enzyme cascade activates other inactive precursors (zymogens). When activated, they lead to target cell lysis and facilitate phagocytosis through opsonization [14,107]. Complements act as key mediators of several pathophysiological processes, and extreme complement activation has a pivotal role in the pathogenesis of several diseases, including paroxysmal nocturnal hemoglobinuria (PNH) and atypical hemolytic uremic syndrome (aHUS) [108].

Eculizumab, an antibody against the complement component C5, was first introduced in 2004 in a preliminary pilot study of eleven participants, and it became the first complement-specific drug approved by the FDA for the treatment of PNH and aHUS. The therapy results showed that complement-targeted therapeutics are safe and effective and can improve patient’s quality of life. However, eculizumab (Soliris) is one of the most expensive drugs in the world, costing around $400,000 a year and putting enormous pressure on public health systems [109,110,111,112]. Aptamer-based therapeutics can offer several cost-effective treatment options. Researchers used SELEX methodology to develop a specific aptamer for the human complement C5 component. This selection process yielded a 38-mer 2′F RNA anti-C5 aptamer, which inhibits the activity of the human factor C5 of the complement cascade [72]. Subsequently, a 38-mer 2′F RNA aptamer was modified by adding 40 kDa PEG to the 5′ end and 3′-3′-linked deoxythymidine to the 3′ end to produce a complement to the C5 inhibitor ARC1905 with a high affinity for complement C5 [72,107]. At the moment, ARC1905 (an anti-C5 aptamer/avacincaptad pegol sodium) is undergoing phase II and III clinical trials for an intravitreal injection (monotherapy/combination therapy) as a treatment for patients with AMD (ClinicalTrials.gov, #NCT02686658). The avacincaptad pegol intravitreal injection led to a significant reduction in geographic atrophy development in eyes with AMD over 12 months. In the future, aptamer-based C5 inhibitors will be studied for other complement-related diseases [72,113].

### 4.3. Aptamers for Anemia of Chronic Disease

Hepcidin, a small 2.8 kDa peptide, is produced predominantly by hepatocytes and is considered the key mediator for iron homeostasis. Its production is regulated by multiple opposing signals, including systemic iron availability, hepatic iron stores, erythropoietic activity, hypoxia, and inflammatory states. In anemia of chronic disease (ACD) and anemia of inflammation, serum hepcidin concentrations are elevated and play a central role in retaining iron within the mononuclear phagocytic system, thereby leaving inadequate iron for the erythroid progenitor cells and, subsequently, causing ACD [114,115]. NOX-H94 is a structured mirror-image pegylated RNA aptamer that can specifically bind to human hepcidin with a high affinity (K_d_ = 0.65 ± 0.06 nM), thereby antagonizing its role in ACD. This hepcidin inhibitor is able to bind with high-affinity human hepcidin, preventing its binding to ferroportin and the reduction in serum iron [116]. The first clinical trial for the use of NOX-H94 (Spiegelmer^®^ lexaptepid pegol) in single and repeated intravenous and subcutaneous administration in healthy subjects showed that it is generally safe and well tolerated, with mild and transient increases in transaminase, serum iron concentration, and transferrin in proportion to the dose [117].

## 5. Conclusions

As the aptamers exhibit a high binding specificity and affinity and have several superior advantages over antibodies, they have become excellent alternatives to antibodies in the diagnostics and therapeutics of blood diseases (Table 1). Aptamers offer a set of tools for novel diagnostics, drug delivery, and therapeutics to treat several types of diseases. Since the first aptamer was discovered, several aptamers have been studied for many biomedical applications. The antivascular endothelial growth factor oligonucleotide-aptamer (NX1838) was the first aptamer to reach human clinical testing in the study titled “Vascular Endothelial Growth Factor (VEGF) as a Therapeutic Target”, and NX1838 (pegaptanib sodium), a nuclease-resistant aptamer, has been designated for use as an ophthalmology injection and was approved by the FDA in December 2004. This approval provided motivation to advance novel aptamer-based therapeutics, unfortunately, no other aptamers have been approved for clinical use since that approval. However, several aptamers are in proof-of-concept studies and various stages of clinical trials, including aptamers for blood diseases. This review presented the relevant research from recent years on aptamers for blood diseases (Figure 3).

## 6. Future Perspectives

Attention to therapeutic aptamers is dramatically increasing year by year. The aptamer-based therapeutics’ main limitations appear to be their rapid degradation by nucleases in the blood and their rapid excretion through renal clearance. Interestingly, this is not the case for limitations in aptamer-based diagnostics, which have fewer limitations and no direct health risks. Some modifications for aptamers have been accomplished, and addressing the aptamer stability may offer more versatile processes for generating aptamers suitable for in vivo use. Aptamers show beneficial therapeutic effects for some blood diseases, and establishing personalized molecular medicine will allow the development of a new generation of targeted therapeutics using agents such as aptamers. However, significant research is still needed to accelerate aptamers’ entry into clinical trials and their clinical use approval.

## Figures and Tables

**Figure 1 molecules-27-00383-f001:**
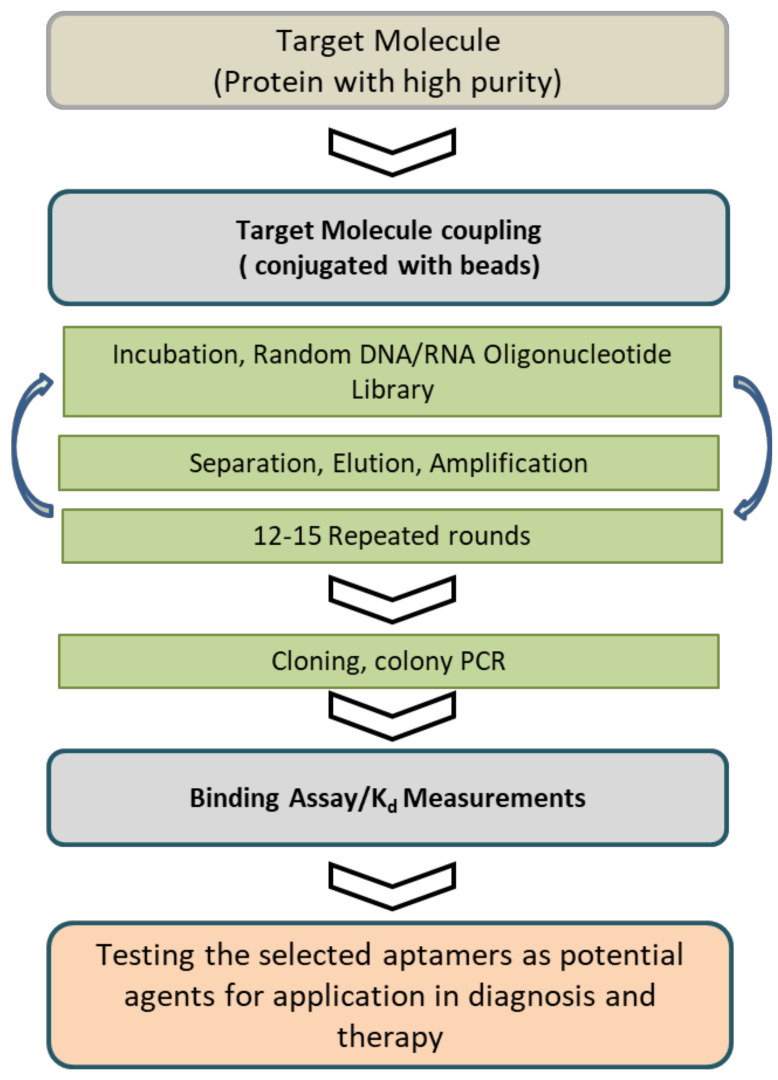
Schematic representation of the protein-based SELEX process used to select an aptamer. The conventional SELEX method is typically carried out using purified target molecules and includes incubating the target molecule. This cycle has to be performed in several rounds, and the binding affinity is monitored until significant binding affinity is reached. High-affinity aptamers for the ligands are isolated and identified by classical cloning and the sequencing results using bioinformatics analysis. The aptamer selection process is followed by selectivity study to the specific target molecule, and the thus identified molecules are used for potential diagnostic or therapeutic applications.

**Figure 2 molecules-27-00383-f002:**
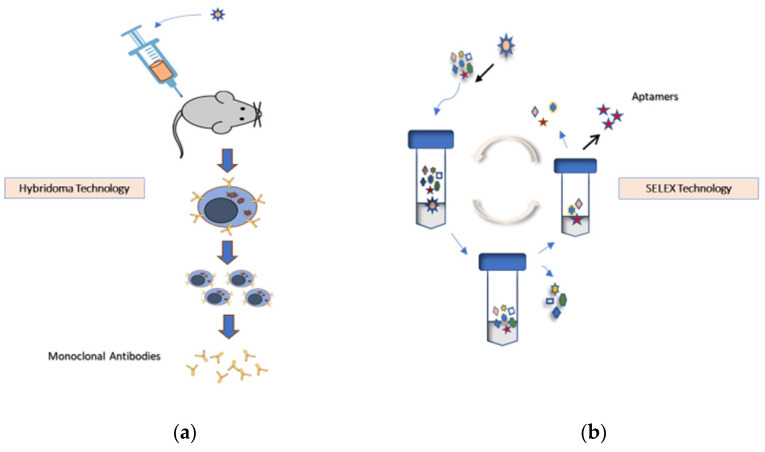
(**a**) Hybridoma technology is the fundamental method for producing identical antibodies (known as mAbs) specific to antigens of interest. It involves injecting animals (usually mice or rats) with an antigen that provokes an immune response for mAb production. This approach presents ethical considerations and restrictions related to infectious risks associated with animal use. Recombinant mAb technology is an essential alternative to using animals for mAb generation and production [2]. (**b**) SELEX technology selects specific aptamers from random DNA, RNA, or peptide libraries without sacrificing animals.

**Figure 3 molecules-27-00383-f003:**
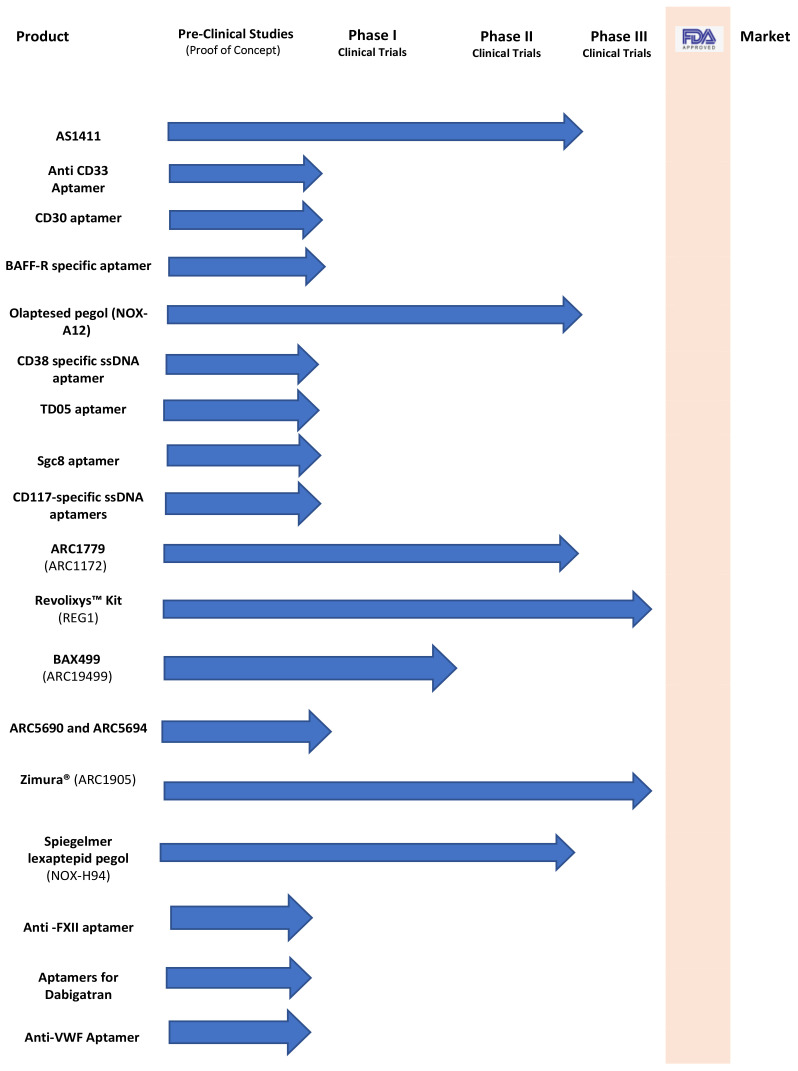
Aptamers in the therapeutics and diagnostics pipeline for blood diseases.

**Table 1 molecules-27-00383-t001:** Comparison of the critical properties of antibodies and aptamers.

Monoclonal Antibodies	Aptamers
Large molecule (IGG monoclonal antibody approximately 150 kDa)	Small molecule (10–100 times smaller than antibodies)
Produced biologically (in vivo) in animal house facilities or reactors	Produced chemically (in vitro)
High cost of synthesis	Low cost of synthesis, large-scale bulk production
Widely distributed technologies (widely used)	Limited distribution of technologies
Difficult to modify	Easy to modify by simple bioconjugate chemistry
Contamination by viral or bacterial during manufacturing process can affect product quality	Chemical/lab manufacturing process carries no risk of biological contamination
High batch-to-batch variation	Low batch-to-batch variation
Clonal variation	No clonal variation
Long half-life in vivo (less susceptible to serum degradation and renal filtration)	Short half-life in vivo (susceptible to serum degradation and renal filtration)
Often immunogenic	Less immunogenic
Thermally unstable	Thermally stable
Limited shelf life	Long shelf life
Poor internalization into cells/tumors	Efficient cellular internalization
Less susceptible to nuclease degradation, rapid elimination from plasma by renal filtration	Susceptible to nuclease degradation, rapid elimination from plasma by renal filtration
Antibody conjugation with one type of signaling or binding molecule, such as organic dyes, fluorescent proteins, colored particles, or enzymes, is typically achieved after antibody formation.	Aptamers can be easily conjugated to different secondary reagents such as small nanoparticles, chemotherapeutic drugs, toxins, enzymes, radionuclides, small interfering RNAs and microRNAs, etc., often during aptamer synthesis, secondary reagents. Conjugation can be readily introduced during synthesis
Limited ability to utilize negative selection pressure	Ability for a counter (negative) selection

**Table 2 molecules-27-00383-t002:** Summary of different pre-clinical and clinical-stage aptamers for blood diseases.

Target	Name of Aptamer	Type of Aptamer	Hematological Indication	Phase of Testing	References
Nucleolin	AS1411 aptamer (AGRO100, later renamed AS1411 then ACT-GRO-777)	26-ntG-rich sequence pegylated DNA aptamer	Acute myeloid leukemia (AML)	Phase II clinical trials in a combination therapy for patients with myeloid leukemia; ClinicalTrials.gov, #NCT00512083 (completed); Phase II clinical trials in a combination therapy for patients with primary refractory or relapsed AML, ClinicalTrials.gov, #NCT01034410 (terminated)	[42]
CD33; transmembrane protein	Anti-CD33 aptamer	DNA aptamer	AML	Pre-clinical studies (proof of concept) are binding and being internalized into CD33-positive myeloid cell lines, carrier of chemotherapeutic drugs	[43]
CD30; transmembrane protein	Anti-CD30 aptamer (C2NP and PS1NP/ truncated form, PS1NPD)	ssDNA aptamer	Hodgkin’s lymphoma (HL) tumor cells	Pre-clinical studies (proof of concept)	[44,45]
B-cell activating factor receptor (BAFF-R)	BAFF-R-specific aptamer	RNA aptamers a	BAFF-R-positive lymphoma cells, such as non-Hodgkin’s lymphoma (NHL)	Pre-clinical studies (proof of concept); specificity of this aptamer to bind and internalize to BAFF-R-positive lymphoma cells, carrier of chemotherapeutic drugs	[46,47]
Stromal cell-derived factor-1 (SDF-1/CXCL 12)	NOX-A12 (olaptesed pegol)	45-nt RNA, L-ribonucleic acid, Spiegelmer, pegylated	Multiple myeloma (MM), CLL	Phase II clinical trials in a combination therapy for MM, ClinicalTrials.gov, #NCT01521533; Phase II clinical trials in a combination therapy for CLL, ClinicalTrials.gov, #NCT01486797	[48,49,50,51,52]
CD38; transmembrane glycoprotein, myeloma cells	CD38-specific ssDNA aptamer	ssDNA	MM	Pre-clinical studies (proof of concept); conjugated to a cytotoxic agent to target and release this agent within MM cells and induce MM cell apoptosis	[53]
Immunoglobulin heavy mu chain (IGHM)	TD05 aptamer	ssDNA	Burkitt’s lymphoma	Pre-clinical studies (proof of concept); successfully recognize tumor cells with high sensitivity and specificity	[54]
Membrane receptor, protein tyrosine kinase 7 (PTK7) tyrosine	Sgc8 aptamer	ssDNA	T-cell acute lymphoblastic leukemia (T-ALL)	Pre-clinical studies (proof of concept); sgc8 aptamer–DOX conjugate possesses high binding affinity and the ability to be efficiently internalized by target cells; “Targeted drug delivery to PTK7 T-ALL leukemia”	[55,56,57,58]
CD117 (c-Kit), transmembrane receptor	CD117-specific ssDNA aptamers	ssDNA	AML	Proof of concept; this Apt-MTX used to target AML cells shows selective growth inhibition of leukemia cells and had no toxicity to normal marrow cells, potential clinical value for use in AML	[59,60,61]
von Willebrand factor (VWF) A1 domain to platelet GPIb receptors	ARC1779	ARC1779 A, pegylated form of DNA aptamer (ARC1172)	von Willebrand factor-related platelet function disorders; Thrombotic thrombocytopenic purpura (TTP) and von Willebrand disease type 2B	Phase II clinical trials for cerebral microembolism in patients undergoing carotid endarterectomy, ClinicalTrials.gov, #NCT00742612 (terminated because of cessation of funding); Phase II for patients with acute TTP, ClinicalTrials.gov, #NCT00726544 (prematurely closed)	[62,63,64]
Factor IXa inhibitor; pegnivacogin (RB006) anivamersen (RB007), complementary active control agent	REG1 System/ Revolixys Kit; pegnivacogin (RB006) plus anivamersen (RB007), a complementary (antisense)	System consists of two RNA aptamers: 1-RB006: 2′-ribo purine/2′- fluoro pyrimidine; 2-RB007: 40 kDa PEG plus 2′-O- methyl antidote	Antithrombotic drug	Phase III clinical trial, ClinicalTrials.gov, #NCT01848106 (clinical hold because of allergic reactions)	ClinicalTrials.gov (accessed on 20 May 2021)
Tissue factor (TF)	BAX499/ ARC19499	-32 nucleotides capping with a 3′ inverted dT—a 5′ 40 kDa PEG	Hemophilia	Phase 1; first-in-human and proof-of-mechanism study in hemophilia patients, ClinicalTrials.gov, #NCT01191372 (terminated)	[65,66,67,68,69]
P-selectin, cell adhesion molecule	Anti-P-selectin aptamers: ARC5690 and ARC5694	ARC5690: -33-mer oligonucleotide- a 3′-inverted 2′-deoxy-thymidine—a 5′-40 kDa PEG	Sickle cell disease (SCD)	Pre-clinical studies (proof of concept); Aptamer base p-selectin increases RBC velocity and wall shear rates and reduces the adhesion of RBCs and leukocytes in SCD model mice	[70,71]
Complement C5	C5-specific aptamer (ARC1905, also known as Zimura)	38-mer 2′F RNA aptamer, 40 kDa PEG to the 5′ end and 3′-3′ linked deoxythymidine to the 3′ end	Potential for complement-related diseases, such as paroxysmal nocturnal hemoglobinuria (PNH)	Phase II/III clinical trials in age-related macular degeneration (AMD) ClinicalTrials.gov, #NCT02686658 Proof of concept of aptamer-based C5 binding and inhibitory activity Potential for study in other complement-related diseases, such as PNH	[72]
Hepcidin peptide	NOX-H94, (Spiegelmer^®^ lexaptepid pegol)	44-nt RNA L-ribonucleic acid, pegylated	Anemia of chronic disease (ACD)	Phase II clinical trials, ClinicalTrials.gov, #NCT02079896 ClinicalTrials.gov, #NCT01691040 ClinicalTrials.gov, #NCT01522794 ClinicalTrials.gov, #NCT01372137	ClinicalTrials.gov (accessed on 20 May 2021)
Human FXII	Anti-FXII aptamer (Aptamer R4cXII 1t)	RNA Aptamer	Thrombosis	Pre-clinical studies (proof of concept); inhibits the intrinsic pathway of coagulation	[73]
Direct oral anticoagulants (dabigatran)	DGB-1, DBG-2, DBG-4, and DBG-5	ssDNA aptamers	Direct oral anticoagulants (dabigatran)	Pre-clinical studies (proof of concept) for monitoring direct oral anticoagulants (dabigatran)	[74,75]
Refrigerated platelets (for platelet transfusion)	Anti-VWF aptamer	von Willebrand factor	Refrigerated platelets (for platelet transfusion)	Pre-clinical studies (proof of concept) for the use of ARC1779 to refrigerated platelets; improves post-transfusion recovery and preserves the long-term hemostatic function of refrigerated platelets	[76]
Serum and plasma in aplastic anemia (AA), noncellular compartment of human bone marrow in AML	SOMAscan proteomic analysis	RNA-sequencing and proteomics data set	AA, AML	Proof of concept to determine the true proteomic of serum and plasma in AA patients before and after therapy Proof of concept to determine the true proteomic composition of the extracellular soluble compartment of AML patients’ bone marrow	[77,78]

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
