# Peer review of "Aptamers: Potential Diagnostic and Therapeutic Agents for Blood Diseases"

_molecules, 2022, doi:10.3390/molecules27020383_

Round 1
Reviewer 1 Report
In this manuscript, the authors summarize and evaluate the importance of aptamers for the diagnosis and therapy of blood diseases in the context of a comprehensive review.
Since the use of aptamers in medicine, especially in the treatment of hematological diseases, is still of little interest, such a review is of particular importance. The advantages of the use of the novel biomolecules are comprehensively presented by the authors. The summary of the studies currently performed is successful. The selection of literature on aptamers is in line with the requirements of a review article.
Suggestions for improvement:
- the manuscript is too long in its present form . Large tables (table 2) should be significantly shortened and moved in their entirety to the appendix. In addition, the formatting should be revised.
- It would be beneficial if the authors focused on a group of hematologic disorders. For example, the primary presentation of aptamers with respect to diagnosis and therapy of leukemias and lymphomas would be sufficient. In its current form, the manuscript is confusing.
- the authors use the term aptamer too often. In one small paragraph for example, the word is used nine times, among others.
- the importance of chemical protection of an aptamer after injection should be emphasized.
Author Response
plz find attached file

Reviewer 2 Report
Review attached.

Author Response
plz find attached file

Round 2
Reviewer 2 Report
I don't understand well what happened, probably the authors uploaded a wrong version of their revised manuscript, since they declared in their response to this Reviewer that they had modified their manuscript according to the suggestions/criticisms, but indeed I could not find the requested revisions in the new version of the manuscript. You can easily check that, for instance, the number and kind of the references are identical in the first and in second version, whereas a substantial revision of the cited literature had been required by this Reviewer (with addition of new articles to be cited and discussed!) and was not carried out.
So, I submit the same comments presented before. My opinion is that the manuscript is still very far from a form which deserves publication.
